# Characteristics and Applications of Technology-Aided Hand Functional Assessment: A Systematic Review

**DOI:** 10.3390/s22010199

**Published:** 2021-12-28

**Authors:** Ciro Mennella, Susanna Alloisio, Antonio Novellino, Federica Viti

**Affiliations:** 1Institute of Biophysics, National Research Council, Via De Marini 6, 16149 Genova, Italy; susanna.alloisio@ettsolutions.com (S.A.); federica.viti@ibf.cnr.it (F.V.); 2ETT Spa, Via Sestri 37, 16154 Genova, Italy; antonio.novellino@ettsolutions.com

**Keywords:** hand, functional assessment, quantitative assessment, kinematic analysis, kinetic analysis, robotic technology, sensing technology

## Abstract

Technology-aided hand functional assessment has received considerable attention in recent years. Its applications are required to obtain objective, reliable, and sensitive methods for clinical decision making. This systematic review aims to investigate and discuss characteristics of technology-aided hand functional assessment and their applications, in terms of the adopted sensing technology, evaluation methods and purposes. Based on the shortcomings of current applications, and opportunities offered by emerging systems, this review aims to support the design and the translation to clinical practice of technology-aided hand functional assessment. To this end, a systematic literature search was led, according to recommended PRISMA guidelines, in PubMed and IEEE Xplore databases. The search yielded 208 records, resulting into 23 articles included in the study. Glove-based systems, instrumented objects and body-networked sensor systems appeared from the search, together with vision-based motion capture systems, end-effector, and exoskeleton systems. Inertial measurement unit (IMU) and force sensing resistor (FSR) resulted the sensing technologies most used for kinematic and kinetic analysis. A lack of standardization in system metrics and assessment methods emerged. Future studies that pertinently discuss the pathophysiological content and clinimetrics properties of new systems are required for leading technologies to clinical acceptance.

## 1. Introduction

The complex anatomy of the hand is efficiently organized to carry out a variety of complex tasks required in every daily activity. Adopting McPhee words, “hand use is a function of anatomic integrity, mobility, strength, sensation, coordination, age, sex, mental status, disease or trauma” [1]. Several injuries and disorders may undermine the physiological hand function, causing severe to subtle functional consequences in day-to-day activities. Pathological events, such as traumatic injuries, rheumatic diseases, metabolic disorders, neurological diseases, neuropsychiatric disorders may undermine hand ability to carry out successfully daily activities, with serious impact on social participation and health-related quality of life [2,3].

Conventional methods for hand functional assessment rely on subjective evaluations, which result not sensitive enough to detect fine changes in impairments, and potentially introducing bias when attempting to model functional recovery [4,5]. In recent decades, clinicians working in hand surgery and therapy field emphasized the need for instruments able to quantify body functions and activity limitations, in order to enable reliable evaluations of hand impairments and related disabilities [6]. The use of quantitative instruments to assess the impact of hand conditions on its functionality and quality of life is essential for clinical decision-making, monitoring patient progress, and evaluating the effectiveness of treatment. Technology-aided approaches are able, on one side, to emphasize the importance of impairment evaluation during tasks and activities involving movements, while on the other hand, to provide objective and traceable descriptions of upper extremity behavior on continuous scales avoiding ceiling effects [7,8]. Emerging technologies are expected to provide crucial information on upper limb impairments in individuals with mild to moderate disability levels, and promise to accurately identify functionally relevant impairments. A great number of technology-aided approaches for upper limb assessment have been reported in literature in recent years, despite very few of them have been used for hand functional assessment issues. Reviews exist presenting overviews of quantitative assessment instruments applied to a specific cohort of subjects carrying upper limb disorders [9,10,11,12,13]. Nevertheless, to the best of the authors’ knowledge, no published work provides a comprehensive state of the art about quantitative methods based on emerging technologies that focus on hand functional assessment. Here, a systematic review on the state-of-the-art about technology-based methods for quantitative hand functional evaluation is presented, in order to provide inspiration for the development of future systems. The review classifies systems based on their technology, measurements characteristics, evaluation methods, and purposes, to provide an overview on the existing sensing technologies for assessing hand function, and to highlight key features where emerging systems may help in solving current issues in the field.

## 2. Materials and Methods

### 2.1. Literature Search Strategy

A literature search was conducted in PubMed and IEEE Xplore databases. Papers addressing the following aspects were selected: functional assessment, upper extremity (hand or fingers), technology-aided approach. A structured search strategy was performed in each electronic database. The performed query is presented in Table 1.

Additionally, articles satisfying eligibility criteria were found by hand searching and included in the review. Only papers published in refereed journals between January 2010 and June 2021 were considered for this systematic review.

### 2.2. Study Selection Process

This review was carried out according to the Cochrane Collaboration methodology [14]. The article selection process based on the PRISMA guidelines [15] is schematically presented in Figure 1. After removal of duplicates, titles and abstracts of the remaining articles were screened, and the full texts read and selected according to inclusion and exclusion criteria. When the same authors published several studies on the same research initiative, only the most recent ones were retained. Inclusion criteria were: (a) articles must concern sensing technology-based systems applied for hand functional assessment issues; (b) articles must be written in English. Exclusion criteria were: (a) traditional mechanical systems (e.g., goniometer); (b) systems concerning diagnostic tests, imaging, and invasive electromyography techniques used to evaluate the integrity of anatomical structures; (c) quantitative analysis performed in static conditions not requiring activity assessment; (d) sensors technical validation that did not require hand functional assessment procedures; (e) reviews and books.

### 2.3. Data Extraction Process

Data extraction was performed manually. The extracted data included: (a) the sensing technology used in the research paper; (b) the type of system implemented, features about the communication protocols (wireless or wired), the needed calibration, the system technology readiness level (TRL) [16], the feedback modality and the metrics extracted from kinematic and kinetic data; (c) the main evaluation features, the evaluation setting (laboratory, clinical, home), the assessment targets regarding hand functions and population.

Following literature classification, retrieved systems were classified as: (a) glove-based system [17], (b) instrumented object [12], (c) body-networked sensor system when wearable sensor nodes (in smartband or body-mounted sensor) communicate among themselves or with other devices [18], (d) vision-based motion capture system [19], (e) end-effector, and (f) exoskeleton system [20]. When the system provided a feedback modality via haptic, visual, auditory, or virtual reality (VR) during the execution of the task, this information was reported. TRL of each system was assessed by authors following the “Technology Readiness Assessment Guide” [16].

The International Classification of Functioning, Disability and Health (ICF), that provides a comprehensive definition, measurement and policy formulations for health and disability in a consistent and internationally comparable manner, was adopted as reference to categorize existing technology-aided functional assessment approaches [6]. The assessment properties of each system were addressed considering when the activity, considered the ability to execute a task or actions, was evaluated at a singular time point in a structured environment (capacity) or when evaluated in unstructured free-living condition (performance) [6]. As described in De Los Reyes-Guzmán et al. work [10], activities were classified as (a) basic tasks involving a simple hand movement (such as finger flexion/extension, tapping, pinch, hand grasp), (b) functional tasks when the subject was invited to perform a point-to-point movement required in basic daily activity (reaching, grasping, releasing), and (c) real activities of daily living (ADLs), such as drinking, eating, cooking, and dressing.

With the aim to classify and discuss emerging technology-aided hand functional assessment in a general core set of hand conditions, van de Ven-Stevens et al. work [21] was adopted as reference to classify the investigated hand functioning domains in articles. Identified domains concerned “mobility of joint functions”, “muscle power functions”, “fine hand use”, and “hand and arm use”.

## 3. Results

**The initial paper search yielded 208 results.** After reviewing titles and abstracts and duplicates rejection, 71 articles were selected. The application of the inclusion and exclusion criteria led to 18 articles related to technology-aided hand functional assessment. Five additional articles were identified from the successive manual targeted search, leading to a total of 23 articles included in the present systematic review (Figure 1). The data extracted from reviewed articles are summarized in Table 2.

The different sensing technologies that have been used in the selected articles are shown in Figure 2. The 75% of the sensing technologies required calibration procedures whether they implemented wireless or wired communication protocols. Instrumented objects and glove-based systems resulted to be the most frequently adopted solutions, for hand functional assessment, present in 30% and 26% of the total works, respectively. Body-networked sensor systems were used in 4 out of the 23 studies [22,24,29,30], while vision-based motion capture systems were used in 3 out of the 23 studies [25,34,38]. Three works adopted exoskeleton [36,42] or end-effector [26] systems.

The 60% of the systems provide feedback during the assessment procedures via visual [22,27,28,31,37,38,40], acoustic [27,28,37], or haptic [26] feedback, or by adopting virtual reality technology to retrieve the online kinematic rendering of hand activity [23,26,36,39,42,44]. Systems with TRL ≤ 4 (*n*= 7) resulted mainly adopted in laboratory setting, while those with TRL > 5 resulted majorly involved in clinical environments (*n* = 12). Higher TRL were used in home settings (TRL = 6 [22]; TRL7 = [28,44]; TRL9 = [43]). The 78% of the functional evaluations aimed to investigate the execution of tasks in a controlled environment, while 5 out of the 23 papers [22,25,28,29,30] studied the performance of functional tasks/ADLs in free-living conditions. The Table A1 (Appendix A) reports the different system metrics labelled in relation to the original articles. 

In Figure 3, systems are classified according to the type of sensor data (kinematic or kinetic), the type of activity executed, the investigated hand functioning domain.

Three population categories were addressed and reported in Table 3: (1) neurological disease (stroke *n* = 9, spinal cord injury *n* = 3, Parkinson’s disease *n* = 1, multiple sclerosis *n* = 1, chronic inflammatory demyelinating polyneuropathy (CIDP) *n* = 1, cerebral palsy *n* = 1); (2) musculoskeletal impairment (stenosing tenosynovitis n = 1, traumatic injuries *n* = 1); and (3) other conditions (developing infants *n* = 1, healthy subjects *n* = 11). The target functioning domain “fine hand use” resulted explored in 18 works, “hand and arm use” in 13 works, “mobility of joint function” in 10 works, “muscle power function” in 9 works.

## 4. Discussion

This work presents a systematic review of featured technologies developed to support hand functional assessment procedures, described in peer reviewed literature published in the last decade (from 2010 on). A total of 208 publications were screened for eligibility and 23 articles were included in the final assessment.

Several advanced technologies have been developed to solve the relevant application problems of traditional approaches used for hand functional assessment [4,5]. Various sensing technologies, embedded in different systems, are used to capture relevant kinematic and kinetic data. These include IMU, FSR, resistive bend sensor, Kinect sensor (Microsoft Corporation, Redmond, WA, USA) and GoPro camera (GoPro Inc., San Mateo, CA, USA), fiber optical sensor, electrical contacts, Hall effect sensor, magnetometer, piezoresistive pressure sensor, strain gage sensor, conductive electrodes. IMUs and FSRs resulted to be the most commonly sensing technology used, since they yield accurate essential values, are easy to use, and are miniaturized in size. Some new developments on innovative sensing technologies are noteworthy and promising though they have been excluded from the review as they resulted not applied in assessment procedures: stretchable carbon nanotube strain sensor [45], bidirectional triboelectric sensors [46], 3D Printed Optical Sensor [47], and fiber Bragg grating sensors (FBG) [48]. 

Different types of system that involve diverse and complementary technologies are currently used to support hand functional assessment procedures: instrumented objects, glove-based systems, body-networked sensor systems, vision-based motion capture systems, end-effector, and exoskeleton systems.

### 4.1. Instrumented Objects and Glove-Based Systems

Instrumented objects and glove-based systems have been recently developed and applied as functional assessment tools [12,17]. From literature screening, they resulted to be the most frequently adopted systems, as well as those presenting the highest technology readiness levels. Instrumented objects are typically equipped by wireless pressure sensors and IMUs, which provide information to analyze grip force modulation in relation to movement-induced load fluctuation, spatial orientation, and movement acceleration. A glove-based system is a hand-worn device including sensors array, specific electronics for data acquisition/processing, and power supply. Glove-based systems are one of the mostly exploited devices for quantifying movement range, joint velocity (angular/linear), quality of movement through bend sensors and IMUs technologies, sometimes providing the kinematic rendering of the task via visual or VR feedbacks [22,31,37,44]. As reported in a recent work [13], glove-based systems result more suitable for hand functional assessment than other capture-motion systems that require more expensive technologies, specific laboratory settings and advanced post-processing steps. However, such systems need to be customized for each individual in order to ensure optimal sensor position and, although some devices implemented wireless solutions, they are typically wired.

### 4.2. Body-Networked Sensor and Vision-Based Motion Capture Systems

Thanks to the evolution of the technology, the interest to develop compact, lightweight, and comfortable wearable sensors has grown during the last years. In recent articles (from 2017 to 2021), the body-networked sensor systems have been widely proposed for clinical applications, to monitor and assess upper limb during functional tasks. In fact, such systems relying on inertial built-in smart-bands sensors or equipped with sensors mounted at finger/wrist level provide meaningful metrics of real-time movements [49]. During recent years, markerless motion capture systems also obtained attention as instruments able to provide unobtrusive monitoring and accurate assessment of functional movements in real world environment. The Microsoft Kinect sensor was applied in two studies to extract meaningful kinematic features during the execution of some standardized functional tasks [34,38]. Furthermore, a wearable vision-based approach, that adopted a head-mounted GoPro camera to provide egocentric first-person videos, was interestingly exploited to monitor and analyze functional interactions of the hand with objects during activities of daily living [25].

### 4.3. End-Effector and Exoskeleton Systems

Designing a robot to actuate hands or fingers is a significant challenge [50]. In fact, robotic systems were not extensively applied for hand functional evaluations [36,42]. The exoskeleton Armeo^®^Spring [42], that helps to support the weight of the upper and lower arm through a system of springs, was used to record the angles of 6 joints, the position of the hand in space and the grip pressure during some VR tasks (e.g., putting fruit in a shopping cart, wiping a window, or catching moving targets on the screen). The FINGER device was applied to analyze finger motions while subjects are playing a serious game [36] thanks to its mechanical design that facilitates wearability to the back of the hand. Finally, a commercial haptic end-effector device (PhantomOmni or Geomagic Touch, 3D Systems, USA) was particularly useful to quantify arm and hand movements, as well as grip forces during a goal-directed manipulation task (virtual peg insertion test), requiring active lifting of the upper limb against gravity [26].

### 4.4. Impact of Quantitative Measurements on Clinical Practice

Although novel technologies can foster the collection of reliable measurements in several dynamic conditions, the use of assessment tasks and measurements is not standardized. In addition, metrics found in literature that analyze movement features do not follow standardized terminology, and studies analyzing psychometric properties are few. In order to help a deeper understanding about emerging technologies for hand functional assessment, and support future developments, a new systems classification based on technology features and assessment properties is proposed here (Figure 3). The most used technology-aided hand functional assessment systems provide kinematic data during several dynamic tasks. Kinematic assessments are supposed to offer fine-grained and objective outcomes on movement quality and have shown to quantify impairments in various pathological conditions affecting “mobility of joint functions”, “fine hand use”, and “hand and arm use” domains [22,25,31,32,37,41,44]. On the other hand, kinetic analysis showed to be fundamental to address significant information about “muscle power functions” and “fine hand use” domains through the evaluation of basic tasks in neurological diseases which impairments cause severe motor disability [27,30,40,42]. Some authors highlighted the needs to develop systems equipped by more than one sensor, to track both kinematic and kinetic data, thus obtaining metrics that express more detailed outcomes about hand function [24,26,28,36]. Remarkably, Schwarz et al. [24] combined complete kinematic motion analysis and interaction force measurements at the fingertip, giving relevant information about all domains of hand functioning in stroke subjects.

Hand functional impairments related to neurological conditions have been particularly explored by new technologies in recent years. Functional consequences resulted addressed majorly in stroke patients than in spinal cord injury, which has minor prevalence in population [51]. Further research must be extended to other neurological disease affecting hand functioning, such as multiple sclerosis, Parkinson’s disease, cerebral palsy, and peripheral neuropathy [52]. Additional efforts are needed to extend kinematic and kinetic analysis to musculoskeletal injuries (e.g., osteoarthritis, carpal tunnel syndrome, rheumatoid arthritis, stenosing tenosynovitis), often causative of complex hand disability [3]. The exploitation of new technologies-aided hand functional assessment methods in the context of neurodegenerative [53,54] and neurodevelopment disorders [43,55] could provide interesting results. As an example, technologies able to track kinematic information about fine hand use in free-living conditions could support early detection of fine motor impairments in neurodegenerative disorders [56,57]. Moreover, quantitative analysis of reaching patterns, grasping forces and power grip maturation patterns could bring a wealth of knowledge about the subtle fine motor impairments that affect children with cerebral palsy [58], attention deficit hyperactivity disorder (ADHD) [59], and autism spectrum disorder [60].

## 5. Conclusions

Technology-aided clinical assessment procedures represent a promising challenge to optimize healthcare services. Several research studies over the past decade introduced technologies aimed to quantify gross motor movements of shoulder, arm, and forearm. Less attention was paid to relevant issues related to the assessment of hand functioning domains. With this work, the authors aimed to address a comprehensive state-of-the-art of technology-aided hand functional assessment technologies, taking into account both technological requirements and clinical assessment needs. 

Advantages in applications of such technologies have been discussed, with the aim to orient the development and usage of assessment systems. The clinical use is still rather limited. The gradual adoption of novel systems as complementary tools to conventional clinical procedures is desirable, and represents a key issue to optimize diagnosis and treatment outcomes in several pathological condition affecting upper limb functioning.

## Figures and Tables

**Figure 1 sensors-22-00199-f001:**
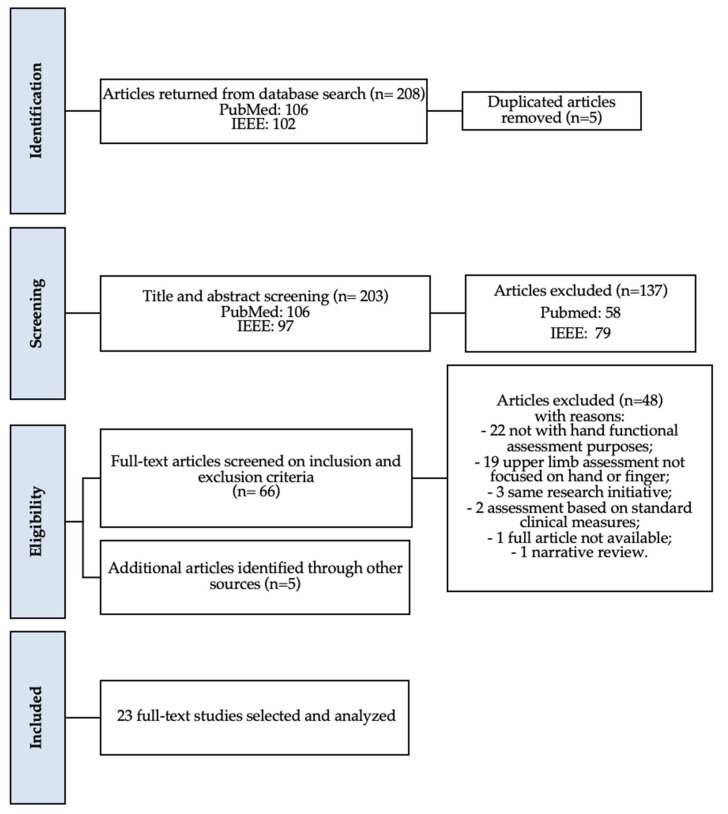
PRISMA flowchart of the results from the literature search.

**Figure 2 sensors-22-00199-f002:**
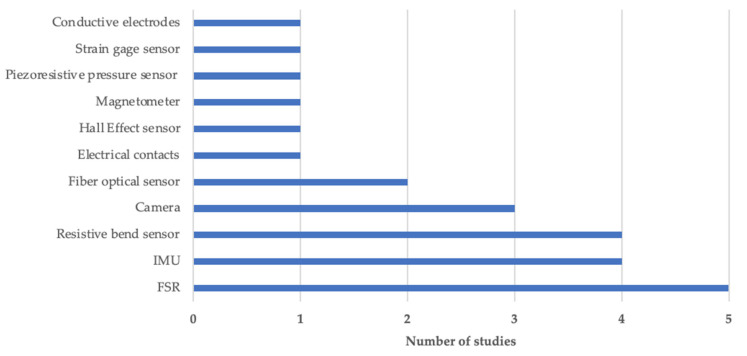
Sensing technology overview. Abbreviations: IMU = Inertial measurement unit, FSR= Force sensing resistor.

**Figure 3 sensors-22-00199-f003:**
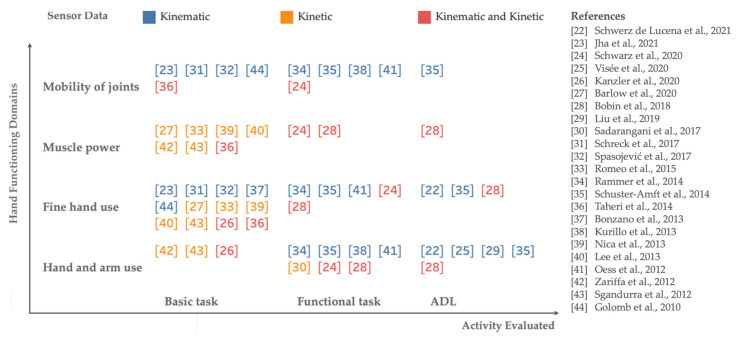
Classification of new systems-aided hand functional assessment based on technology and evaluation features. References: [22] = Schwerz de Lucena et al., 2021; [23] = Jha et al., 2021; [24] = Schwarz et al., 2020; [25] = Visee et al., 2020; [26] = Kanzler et al., 2020; [27] = Barlow et al., 2020; [28] = Bobin et al., 2018; [29] = Liu et al., 2019; [30] = Sadarangani et al., 2017; [31] = Schreck et al., 2017; [32] = Spasojević et al., 2017; [33] = Romeo et al., 2015; [34] = Rammer et al., 2014; [35] = Schuster-Amft et al., 2014; [36] = Taheri et al., 2014; [37] = Bonzano et al., 2013; [38] = Kurillo et al., 2013; [39] = Nica et al., 2013; [40] = Lee et al., 2013; [41] = Oess et al., 2012; [42] = Zariffa et al., 2012; [43] = Sgandurra et al., 2012; [44] = Golomb et al., 2010.

**Table 1 sensors-22-00199-t001:** Literature search strategy.

Concept	Search Terms
Assessment	functional assessment OR monitoring
	AND
Functions/Impairment	range of motion OR muscle power OR fine hand use OR hand activity OR fine impairment
	AND
Upper extremity	upper extremity OR hand OR finger
	AND
Technology-aided approach	technology OR quantitative OR robot OR sensors OR sensor system OR wearable systems OR mobile OR kinematic OR kinetic NOT electromyography

**Table 2 sensors-22-00199-t002:** Summary of the paper lists and features. Ref. = reference, TRL = technology readiness level, IMU = inertial measurement unit, ADL = activities of daily living, Clin = clinical, Lab = laboratory, FSR = force sensing resistor, VR = virtual reality, CIDP = chronic inflammatory demyelinating polyneuropathy.

First Author, Year	Ref.	Sensing Technology	System	Communication Protocols	Calibration	Feedback	Data	Evaluation Type	Activity	Target Functions	Target Population	Setting	TRL
Schwerz de Lucena, 2021	[22]	Magnetometers, IMU	Body- networked sensor system (Wristband and ring)	Wireless	-	Visual	Kinematic	Performance	ADL	Fine hand use; hand and arm use	Chronic stroke (*n* = 29)	Home	TRL 6
Jha, 2021	[23]	Fiber optical sensors	Glove-based system	Wired	Required	VR	Kinematic	Capacity	Basic task	Mobility of joint functions; Fine hand use	Healthy subjects (*n* = 5)	Lab	TRL 4
Schwar, 2020	[24]	Force sensor, IMUs	Body- networked sensor system	Wired	Required	-	Kinematic kinetic	Capacity	Functional task	Mobility of joint functions; Muscle power function; Fine hand use; Hand and arm use	Chronic stroke (*n* = 10)	Clin	TRL 5
Visée, 2020	[25]	GoPro camera sensor	Vision-based motion capture system	Wireless	-	-	Kinematic	Performance	ADL	Hand and arm use	Spinal cord injury (*n* = 17)	Lab	TRL 4
Kanzler, 2020	[26]	Force sensor	End-effector	Wired	-	VR haptic	Kinematic kinetic	Capacity	Basic task	Fine hand use; hand and arm use	Stroke (*n* = 30)	Clin	TRL 7
Barlow, 2020	[27]	Strain gage sensors (bulit-in load cell)	Instrumented object	Wireless	Required	Visual acoustic	Kinetic	Capacity	Basic task	Muscle power functions; Fine hand use	Chronic stroke (n = 7); Healthy subjects (*n* = 25)	Lab	TRL 4
Bobin, 2018	[28]	Pressure sensors (FSR), conductive electrodes, IMU	Instrumented object (Smart cup)	Wireless	Required	Visual acoustic	Kinematic kinetic	Capacity Performance	Functional task; ADL	Muscle power functions; Fine hand use; Hand and arm use	Stroke (*n* = 9)	Clin Home	TRL 7
Liu, 2019	[29]	IMUs	Body-networked sensor system (finger worn sensor, wrist worn sensor)	Wireless	-	-	Kinematic	Performance	ADL	Hand and arm use	Healthy subjects (*n* = 18)	Lab	TRL 4
Sadarangani, 2017	[30]	Force sensors (FSR)	Body-networked sensor system (Smartband)	Wired	Required	-	Kinetic	Performance	Functional task	Hand and arm use	Stroke (*n* = 8); Healthy subjects (*n* = 8)	Lab	TRL 4
Schreck, 2017	[31]	Resistive bend sensors	Glove-based system	Wireless	Required	Visual	Kinematic	Capacity	Basic task	Mobility of joint functions; Fine hand use	Healthy subjects (*n* = 10); Stenosing tenosynovitis (*n* = 11)	Clin	TRL 9
Spasojević, 2017	[32]	Resistive bend sensors	Glove-based system	Wireless	Required	-	Kinematic	Capacity	Basic task	Mobility of joint functions; Fine hand use	Parkinson’s disease (*n* = 30); Healthy subjects (*n* = 23)	Clin	TRL 9
Romeo, 2015	[33]	Force sensor (FSR)	Instrumented object	Wired	Required	-	Kinetic	Capacity	Basic task	Muscle power functions; Fine hand use	Healthy subject (*n* = 1)	Lab	TRL 3
Rammer, 2014	[34]	Microsoft Kinect sensor	Vision-based motion capture system	Wireless	-	-	Kinematic	Capacity	Functional task	Mobility of joint functions; Fine hand use; Hand and arm use	Healthy adolescent subjects (*n* = 12)	Clin	TRL 9
Schuster-Amft, 2014	[35]	Resistive bend sensors	Instrumented object (smart cup)	Wireless	-	VR	Kinematic	Capacity	Functional task	Mobility of joint functions; Fine hand use; Hand and arm use	Chronic stroke (*n* = 60)	Clin	TRL 9
Taheri, 2014	[36]	Hall Effect sensors	Exoskeleton	-	-	VR	Kinematic kinetic	Capacity	Basic task	Mobility of joint functions; Muscle power functions; Fine hand use	Stroke (*n* = 16)	Lab	TRL 4
Bonzano, 2013	[37]	Electrical con- tacts	Glove-based system	Wired	-	Visual acoustic	Kinematic	Capacity	Basic task	Fine hand use	Multiple sclerosis (*n* = 40)	Clin	TRL 8
Kurillo, 2013	[38]	Microsoft Kinect sensor	Vision-based motion capture system	Wireless	Required	Visual	Kinematic	Capacity	Functional task	Mobility of joint functions; Hand and arm use	Healthy subjects (*n* = 10)	Lab	TRL 9
Nica, 2013	[39]	Force sensor	Instrumented object	Wired	-	VR	Kinetic	Capacity	Basic task	Musclepower functions; Fine hand use	Hand traumatic injuries (*n* = 54)	Clin	TRL 9
Lee, 2013	[40]	Force sensor (FSR)	Instrumented object	Wireless	Required	Visual	Kinetic	Capacity	Basic task	Muscle power functions; Fine hand use	Stroke and CIDP (*n* = 12); Healthy subjects (*n* = 4)	Clin	TRL 5
Oess, 2012	[41]	Resistive bend sensors	Glove-based system	Wired	-	-	Kinematic	Capacity	Functional task	Mobility of joint functions; Fine hand use; Hand and arm use	Healthy subjects (*n* = 10); Cervical spine cord injury (*n* = 4)	Clin	TRL 5
Zariffa, 2012	[42]	Pressure sensor	Exoskeleton	Wired	Required	VR	Kinetic	Capacity	Basic task	Muscle power functions; Hand and arm use	Spinal cord injury (*n* = 14)	Clin	TRL 9
Sgandurra, 2012	[43]	Piezoresistive pressure sensor	Instrumented object (ring- shaped toy)	-	-	-	Kinetic	Capacity	Basic task	Muscle power functions; Fine hand use; Hand and arm use	Developing infants from 4-9 months (*n* = 10)	Home	TRL 9
Golomb, 2010	[44]	Fiber optical sensors	Glove-based system	Wired	Required	VR (game)	Kinematic	Capacity	Basic task	Mobility of joint functions; Finehand use	Adolescent with cerebral palsy (*n* = 3)	Home	TRL 7

**Table 3 sensors-22-00199-t003:** Classification based on target population. CIDP = chronic inflammatory demyelinating polyneuropathy.

Category	Target Population	References
Neurological disease	Stroke	[22,24,26,27,28,30,35,36,40]
	Spinal cord injury	[25,41,42]
	Parkinson’s disease	[32]
	Multiple sclerosis	[37]
	CIDP	[40]
	Cerebral palsy	[44]
Musculoskeletal impairment	Stenosing tenosynovitis	[31]
	Traumatic injuries	[39]
Others	Healthy subjects	[23,27,29,30,31,32,33,34,38,40,41]
	Developing infants	[43]

## Data Availability

Not applicable.

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
