# Peer review of "Characteristics and Applications of Technology-Aided Hand Functional Assessment: A Systematic Review"

_sensors, 2021, doi:10.3390/s22010199_

Round 1
Reviewer 1 Report
The authors present a systematic review about the use of technology-aided for hand fuction assessment. Overall, the paper is well written and I enjoy reading the manuscript.
The articicle provide sufficient details about the selection criteria using various databases. I have a small recommendation. The authors should organize the discussion section into subsection for the ease of reading.
To sum up, I believe that this work has a certain contribution to the field and I recommend acceptance.
Author Response
"Please see the attachment."

Reviewer 2 Report
General comments
This manuscript aims at presenting a comprehensive systematic review on the state-of-art about technology-based methods for quantitative hand functional evaluation to provide inspiration for the development of future systems. Such an aim is very noble. Authors highlight a lack of standardization in system metrics and assessment methods emerged. They recommend that duture studies pertinently discuss the pathophysiological content and clinimetrics properties of new systems to lead technologies to clinical acceptance. Overall, authors manage to fulfill properly their aim.
Specific comments
Main questions addressed by the research are which technology-aided hand functional assessment is operated nowadays and which are its applications in terms of aims, used technology and data analysis.
This topic is surely both original and relevant in the field, because new technology became available and therefore allow new applications not feasible neither in the recent past.
This MS has the merit to organize available knowledge in the field providing authors aiming at performing experimental studies with all they need to know to start their work.
Operated methodology is already fine.
Conclusions are consistent with the evidence and the arguments presented and properly answer the initial questions.
References are appropriate and selected bibliography is complete.
Minor comments
(line 41 and elsewhere throughout MS) … 4,5… (viz., extra space)
(l261) … [25]. Designing… (i.e., missing space)
Author Response
"Please see the attachment."
